# Near–Optimal Density Estimation in Near–Linear Time Using Variable–Width Histograms

**Siu-On Chan**
Microsoft Research
sochan@gmail.com

**Ilias Diakonikolas**
University of Edinburgh
ilias.d@ed.ac.uk

**Rocco A. Servedio**
Columbia University
rocco@cs.columbia.edu

**Xiaorui Sun**
Columbia University
xiaoruisun@cs.columbia.edu

## Abstract

Let $p$ be an unknown and arbitrary probability distribution over $[0, 1)$. We consider the problem of *density estimation*, in which a learning algorithm is given i.i.d. draws from $p$ and must (with high probability) output a hypothesis distribution that is close to $p$. The main contribution of this paper is a highly efficient density estimation algorithm for learning using a variable-width histogram, i.e., a hypothesis distribution with a piecewise constant probability density function.

In more detail, for any $k$ and $\varepsilon$, we give an algorithm that makes $\tilde{O}(k/\varepsilon^2)$ draws from $p$, runs in $\tilde{O}(k/\varepsilon^2)$ time, and outputs a hypothesis distribution $h$ that is piecewise constant with $O(k \log^2(1/\varepsilon))$ pieces. With high probability the hypothesis $h$ satisfies $d_{\mathrm{TV}}(p, h) \leq C \cdot \mathrm{opt}_k(p) + \varepsilon$, where $d_{\mathrm{TV}}$ denotes the total variation distance (statistical distance), $C$ is a universal constant, and $\mathrm{opt}_k(p)$ is the smallest total variation distance between $p$ and any $k$-piecewise constant distribution. The sample size and running time of our algorithm are optimal up to logarithmic factors. The "approximation factor" $C$ in our result is inherent in the problem, as we prove that no algorithm with sample size bounded in terms of $k$ and $\varepsilon$ can achieve $C < 2$ regardless of what kind of hypothesis distribution it uses.

## 1 Introduction

Consider the following fundamental statistical task: *Given independent draws from an unknown probability distribution, what is the minimum sample size needed to obtain an accurate estimate of the distribution?* This is the question of *density estimation*, a classical problem in statistics with a rich history and an extensive literature (see e.g., [BBBB72, DG85, Sil86, Sco92, DL01]). While this broad question has mostly been studied from an information–theoretic perspective, it is an inherently algorithmic question as well, since the ultimate goal is to describe and understand algorithms that are both computationally and information-theoretically efficient. The need for computationally efficient learning algorithms is only becoming more acute with the recent flood of data across the sciences; the "gold standard" in this "big data" context is an algorithm with information-theoretically (near-) optimal sample size and running time (near-) linear in its sample size.

In this paper we consider learning scenarios in which an algorithm is given an input data set which is a sample of i.i.d. draws from an unknown probability distribution. It is natural to expect (and can be easily formalized) that, if the underlying distribution of the data is inherently "complex", it may be hard to even approximately reconstruct the distribution. But what if the underlying distribution is "simple" or "succinct" – can we then reconstruct the distribution to high accuracy in a computationally and sample-efficient way? In this paper we answer this question in the affirmative for the

problem of learning "noisy" *histograms*, arguably one of the most basic density estimation problems in the literature.

To motivate our results, we begin by briefly recalling the role of histograms in density estimation. Histograms constitute "the oldest and most widely used method for density estimation" [Sil86], first introduced by Karl Pearson in [Pea95]. Given a sample from a probability density function (pdf) $p$, the method partitions the domain into a number of intervals (bins) $B_1, \ldots, B_k$, and outputs the "empirical" pdf which is constant within each bin. A $k$-*histogram* is a piecewise constant distribution over bins $B_1, \ldots, B_k$, where the probability mass of each interval $B_j$, $j \in [k]$, equals the fraction of observations in the interval. Thus, the goal of the "histogram method" is to approximate an unknown pdf $p$ by an appropriate $k$-histogram. It should be emphasized that the number $k$ of bins to be used and the "width" and location of each bin are unspecified; they are parameters of the estimation problem and are typically selected in an *ad hoc* manner.

We study the following distribution learning question:

> *Suppose that there* exists *a $k$-histogram that provides an accurate approximation to the unknown target distribution. Can we efficiently* find *such an approximation?*

In this paper, we provide a fairly complete affirmative answer to this basic question. Given a bound $k$ on the number of intervals, we give an algorithm that uses a near-optimal sample size, runs in *near-linear time* (in its sample size), and approximates the target distribution nearly as accurately as the best $k$-histogram.

To formally state our main result, we will need a few definitions. We work in a standard model of learning an unknown probability distribution from samples, essentially that of [KMR+94], which is a natural analogue of Valiant's well-known PAC model for learning Boolean functions [Val84] to the unsupervised setting of learning an unknown probability distribution.[1] A distribution learning problem is defined by a class $\mathcal{C}$ of distributions over a domain $\Omega$. The algorithm has access to independent draws from an unknown pdf $p$, and its goal is to output a hypothesis distribution $h$ that is "close" to the target distribution $p$. We measure the closeness between distributions using the *statistical distance* or total variation distance. In the "noiseless" setting, we are promised that $p \in \mathcal{C}$ and the goal is to construct a hypothesis $h$ such that (with high probability) the total variation distance $d_{\mathrm{TV}}(h, p)$ between $h$ and $p$ is at most $\varepsilon$, where $\varepsilon > 0$ is the accuracy parameter.

The more challenging "noisy" or *agnostic* model captures the situation of having arbitrary (or even adversarial) noise in the data. In this setting, we do not make any assumptions about the target density $p$ and the goal is to find a hypothesis $h$ that is almost as accurate as the "best" approximation of $p$ by any distribution in $\mathcal{C}$. Formally, given sample access to a (potentially arbitrary) target distribution $p$ and $\varepsilon > 0$, the goal of an *agnostic learning algorithm for $\mathcal{C}$* is to compute a hypothesis distribution $h$ such that $d_{\mathrm{TV}}(h, p) \leq \alpha \cdot \mathrm{opt}_{\mathcal{C}}(p) + \varepsilon$, where $\mathrm{opt}_{\mathcal{C}}(p) := \inf_{q \in \mathcal{C}} d_{\mathrm{TV}}(q, p)$ – i.e., $\mathrm{opt}_{\mathcal{C}}(p)$ is the statistical distance between $p$ and the closest distribution to it in $\mathcal{C}$ – and $\alpha \geq 1$ is a constant (that may depend on the class $\mathcal{C}$). We will call such a learning algorithm an *$\alpha$-agnostic learning algorithm for $\mathcal{C}$*; when $\alpha > 1$ we sometimes refer to this as a *semi-agnostic learning algorithm*.

A distribution $f$ over a finite interval $I \subseteq \mathbb{R}$ is called $k$-*flat* if there exists a partition of $I$ into $k$ intervals $I_1, \ldots, I_k$ such that the pdf $f$ is constant within each such interval. We henceforth (without loss of generality for densities with bounded support) restrict ourselves to the case $I = [0, 1)$. Let $\mathcal{C}_k$ be the class of all $k$-*flat* distributions over $[0, 1)$. For a (potentially arbitrary) distribution $p$ over $[0, 1)$ we will denote by $\mathrm{opt}_k(p) := \inf_{f \in \mathcal{C}_k} d_{\mathrm{TV}}(f, p)$.

In this terminology, our learning problem is exactly the problem of agnostically learning the class of $k$-flat distributions. Our main positive result is a near-optimal algorithm for this problem, i.e., a semi-agnostic learning algorithm that has near-optimal sample size and near-linear running time. More precisely, we prove the following:

**Theorem 1** (Main). *There is an algorithm $A$ with the following property: Given $k \geq 1$, $\varepsilon > 0$, and sample access to a target distribution $p$, algorithm $A$ uses $\tilde{O}(k/\varepsilon^2)$ independent draws from $p$, runs in time $\tilde{O}(k/\varepsilon^2)$, and outputs a $O(k \log^2(1/\varepsilon))$-flat hypothesis distribution $h$ that satisfies $d_{\mathrm{TV}}(h, p) \leq O(\mathrm{opt}_k(p)) + \varepsilon$ with probability at least $9/10$.*

Using standard techniques, the confidence probability can be boosted to $1 - \delta$, for any $\delta > 0$, with a (necessary) overhead of $O(\log(1/\delta))$ in the sample size and the running time.

We emphasize that the difficulty of our result lies in the fact that the "optimal" piecewise constant decomposition of the domain is both *unknown* and *approximate* (in the sense that $\mathrm{opt}_k(p) > 0$); and that our algorithm is both sample-optimal and runs in (near-) *linear time*. Even in the (significantly easier) case that the target $p \in \mathcal{C}_k$ (i.e., $\mathrm{opt}_k(p) = 0$), and the optimal partition is explicitly given to the algorithm, it is known that a sample of size $\Omega(k/\varepsilon^2)$ is information-theoretically necessary. (This lower bound can, e.g., be deduced from the standard fact that learning an unknown discrete distribution over a $k$-element set to statistical distance $\varepsilon$ requires an $\Omega(k/\varepsilon^2)$ size sample.) Hence, our algorithm has provably optimal sample complexity (up to a logarithmic factor), runs in essentially sample linear time, and is $\alpha$-agnostic for a universal constant $\alpha > 1$.

It should be noted that the sample size required for our problem is well-understood; it follows from the VC theorem (Theorem 3) that $O(k/\varepsilon^2)$ draws from $p$ are information-theoretically sufficient. However, the theorem is non-constructive, and the "obvious" algorithm following from it has running time exponential in $k$ and $1/\varepsilon$. In recent work, Chan *et al* [CDSS14] presented an approach employing an intricate combination of dynamic programming and linear programming which yields a $\mathrm{poly}(k/\varepsilon)$ time algorithm for the above problem. However, the running time of the [CDSS14] algorithm is $\Omega(k^3)$ even for constant values of $\varepsilon$, making it impractical for applications. As discussed below our algorithmic approach is significantly different from that of [CDSS14], using neither dynamic nor linear programming.

**Applications.** Nonparametric density estimation for shape restricted classes has been a subject of study in statistics since the 1950's (see [BBBB72] for an early book on the topic and [Gre56, Bru58, Rao69, Weg70, HP76, Gro85, Bir87] for some of the early literature), and has applications to a range of areas including reliability theory (see [Reb05] and references therein). By using the structural approximation results of Chan *et al* [CDSS13], as an immediate corollary of Theorem 1 we obtain sample optimal and *near-linear time* estimators for various well-studied classes of shape restricted densities including monotone, unimodal, and multimodal densities (with unknown mode locations), monotone hazard rate (MHR) distributions, and others (because of space constraints we do not enumerate the exact descriptions of these classes or statements of these results here, but instead refer the interested reader to [CDSS13]). Birgé [Bir87] obtained a sample optimal and linear time estimator for monotone densities, but prior to our work, no linear time and sample optimal estimator was known for any of the other classes.

Our algorithm from Theorem 1 is $\alpha$-agnostic for a constant $\alpha > 1$. It is natural to ask whether a significantly stronger accuracy guarantee is efficiently achievable; in particular, is there an agnostic algorithm with similar running time and sample complexity and $\alpha = 1$? Perhaps surprisingly, we provide a negative answer to this question. Even in the simplest nontrivial case that $k = 2$, and the target distribution is defined over a discrete domain $[N] = \{1, \ldots, N\}$, any $\alpha$-agnostic algorithm with $\alpha < 2$ requires large sample size:

**Theorem 2** (Lower bound, Informal statement). *Any* $1.99$-*agnostic learning algorithm for* $2$-*flat distributions over* $[N]$ *requires a sample of size* $\Omega(\sqrt{N})$.

See Theorem 7 in Section 4 for a precise statement. Note that there is an exact correspondence between distributions over the discrete domain $[N]$ and pdf's over $[0, 1)$ which are piecewise constant on each interval of the form $[k/N, (k + 1)/N)$ for $k \in \{0, 1, \ldots, N - 1\}$. Thus, Theorem 2 implies that *no finite sample* algorithm can $1.99$-agnostically learn even $2$-flat distributions over $[0, 1)$. (See Corollary 4.1 in Section 4 for a detailed statement.)

**Related work.** A number of techniques for density estimation have been developed in the mathematical statistics literature, including kernels and variants thereof, nearest neighbor estimators, orthogonal series estimators, maximum likelihood estimators (MLE), and others (see Chapter 2 of [Sil86] for a survey of existing methods). The main focus of these methods has been on the statistical rate of convergence, as opposed to the running time of the corresponding estimators. We remark that the MLE does not exist for very simple classes of distributions (e.g., unimodal distributions with an unknown mode, see e.g, [Bir97]). We note that the notion of agnostic learning is related to the literature on model selection and oracle inequalities [MP007], however this work is of a different flavor and is not technically related to our results.

Histograms have also been studied extensively in various areas of computer science, including databases and streaming [JKM$^+$98, GKS06, CMN98, GGI$^+$02] under various assumptions about the input data and the precise objective. Recently, Indyk *et al* [ILR12] studied the problem of learning a $k$-flat distribution over $[N]$ *under the $L_2$ norm* and gave an efficient algorithm with sample complexity $O(k^2 \log(N)/\varepsilon^4)$. Since the $L_1$ distance is a stronger metric, Theorem 1 implies an improved sample and time bound of $\tilde{O}(k/\varepsilon^2)$ for their setting.

## 2 Preliminaries

Throughout the paper we assume that the underlying distributions have Lebesgue measurable densities. For a pdf $p : [0,1) \to \mathbb{R}_+$ and a Lebesgue measurable subset $A \subseteq [0,1)$, i.e., $A \in \mathcal{L}([0,1))$, we use $p(A)$ to denote $\int_{z \in A} p(z)$. The *statistical distance* or *total variation distance* between two densities $p, q : [0,1) \to \mathbb{R}_+$ is $d_{\text{TV}}(p,q) := \sup_{A \in \mathcal{L}([0,1))} |p(A) - q(A)|$. The statistical distance satisfies the identity $d_{\text{TV}}(p,q) = \frac{1}{2} \|p - q\|_1$ where $\|p - q\|_1$, the $L_1$ distance between $p$ and $q$, is $\int_{[0,1)} |p(x) - q(x)| dx$; for convenience in the rest of the paper we work with $L_1$ distance. We refer to a nonnegative function $p$ over an interval (which need not necessarily integrate to one over the interval) as a "sub-distribution." Given a value $\kappa > 0$, we say that a (sub-)distribution $p$ over $[0,1)$ is $\kappa$-*well-behaved* if $\sup_{x \in [0,1)} \mathbf{Pr}_{x \sim p}[x] \leq \kappa$, i.e., no individual real value is assigned more than $\kappa$ probability under $p$. Any probability distribution with no atoms is $\kappa$-well-behaved for all $\kappa > 0$. Our results apply for general distributions over $[0,1)$ which may have an atomic part as well as a non-atomic part. Given $m$ independent draws $s_1, \ldots, s_m$ from a distribution $p$ over $[0,1)$, the *empirical distribution* $\widehat{p}_m$ over $[0,1)$ is the discrete distribution supported on $\{s_1, \ldots, s_m\}$ defined as follows: for all $z \in [0,1)$, $\mathbf{Pr}_{x \sim \widehat{p}_m}[x = z] = |\{j \in [m] \mid s_j = z\}|/m$.

**The VC inequality.** Let $p : [0,1) \to \mathbb{R}$ be a Lebesgue measurable function. Given a family of subsets $\mathcal{A} \subseteq \mathcal{L}([0,1))$ over $[0,1)$, define $\|p\|_{\mathcal{A}} = \sup_{A \in \mathcal{A}} |p(A)|$. The *VC dimension* of $\mathcal{A}$ is the maximum size of a subset $X \subseteq [0,1)$ that is shattered by $\mathcal{A}$ (a set $X$ is shattered by $\mathcal{A}$ if for every $Y \subseteq X$, some $A \in \mathcal{A}$ satisfies $A \cap X = Y$). If there is a shattered subset of size $s$ for all $s \in \mathbb{Z}_+$, then we say that the VC dimension of $\mathcal{A}$ is $\infty$. The well-known *Vapnik-Chervonenkis (VC) inequality* states the following:

**Theorem 3** (VC inequality, [DL01, p.31])**.** *Let $p : I \to \mathbb{R}_+$ be a probability density function over $I \subseteq \mathbb{R}$ and $\widehat{p}_m$ be the empirical distribution obtained after drawing $m$ points from $p$. Let $\mathcal{A} \subseteq 2^I$ be a family of subsets with VC dimension $d$. Then $\mathbf{E}[\|p - \widehat{p}_m\|_{\mathcal{A}}] \leq O(\sqrt{d/m})$.*

**Partitioning into intervals of approximately equal mass.** As a basic primitive, given access to a sample drawn from a $\kappa$-well-behaved target distribution $p$ over $[0,1)$, we will need to partition $[0,1)$ into $\Theta(1/\kappa)$ intervals each of which has probability $\Theta(\kappa)$ under $p$. There is a simple algorithm, based on order statistics, which does this and has the following performance guarantee (see Appendix A.2 of [CDSS14]):

**Lemma 2.1.** *Given $\kappa \in (0,1)$ and access to points drawn from a $\kappa/64$-well-behaved distribution $p$ over $[0,1)$, the procedure* Approximately-Equal-Partition *draws $O((1/\kappa)\log(1/\kappa))$ points from $p$, runs in time $\tilde{O}(1/\kappa)$, and with probability at least $99/100$ outputs a partition of $[0,1)$ into $\ell = \Theta(1/\kappa)$ intervals such that $p(I_j) \in [\kappa/2, 3\kappa]$ for all $1 \leq j \leq \ell$.*

## 3 The algorithm and its analysis

In this section we prove our main algorithmic result, Theorem 1. Our approach has the following high-level structure: In Section 3.1 we give an algorithm for agnostically learning a target distribution $p$ that is "nice" in two senses: (i) $p$ is well-behaved (i.e., it does not have any heavy atomic elements), and (ii) $\text{opt}_k(p)$ is bounded from above by the error parameter $\varepsilon$. In Section 3.2 we give a general efficient reduction showing how the second assumption can be removed, and in Section 3.3 we briefly explain how the first assumption can be removed, thus yielding Theorem 1.

### 3.1 The main algorithm

In this section we give our main algorithmic result, which handles well-behaved distributions $p$ for which $\mathrm{opt}_k(p)$ is not too large:

**Theorem 4.** *There is an algorithm* Learn-WB-small-opt-$k$-histogram *that given as input* $\tilde{O}(k/\varepsilon^2)$ *i.i.d. draws from a target distribution* $p$ *and a parameter* $\varepsilon > 0$, *runs in time* $\tilde{O}(k/\varepsilon^2)$, *and has the following performance guarantee: If (i)* $p$ *is* $\frac{\varepsilon/\log(1/\varepsilon)}{384k}$-*well-behaved, and (ii)* $\mathrm{opt}_k(p) \leq \varepsilon$, *then with probability at least* $19/20$, *it outputs an* $O(k \cdot \log^2(1/\varepsilon))$-*flat distribution* $h$ *such that* $d_{\mathrm{TV}}(p, h) \leq 2 \cdot \mathrm{opt}_k(p) + 3\varepsilon$.

We require some notation and terminology. Let $r$ be a distribution over $[0, 1)$, and let $\mathcal{P}$ be a set of disjoint intervals that are contained in $[0, 1)$. We say that the $\mathcal{P}$-*flattening of* $r$, denoted $(r)^{\mathcal{P}}$, is the sub-distribution defined as

$$r(v) = \begin{cases} r(I)/|I| & \text{if } v \in I, I \in \mathcal{P} \\ 0 & \text{if } v \text{ does not belong to any } I \in \mathcal{P} \end{cases}$$

Observe that if $\mathcal{P}$ is a partition of $[0, 1)$, then (since $r$ is a distribution) $(r)^{\mathcal{P}}$ is a distribution.

We say that two intervals $I, I'$ are *consecutive* if $I = [a, b)$ and $I' = [b, c)$. Given two consecutive intervals $I, I'$ contained in $[0, 1)$ and a sub-distribution $r$, we use $\alpha_r(I, I')$ to denote the $L_1$ distance between $(r)^{\{I, I'\}}$ and $(r)^{\{I \cup I'\}}$, i.e., $\alpha_r(I, I') = \int_{I \cup I'} |(r)^{\{I, I'\}}(x) - (r)^{\{I \cup I'\}}(x)| dx$. Note here that $\{I \cup I'\}$ is a set that contains one element, the interval $[a, c)$.

#### 3.1.1 Intuition for the algorithm

We begin with a high-level intuitive explanation of the Learn-WB-small-opt-$k$-histogram algorithm. It starts in Step 1 by constructing a partition of $[0, 1)$ into $z = \Theta(k/\varepsilon')$ intervals $I_1, \ldots, I_z$ (where $\varepsilon' = \tilde{\Theta}(\varepsilon)$) such that $p$ has weight $\Theta(\varepsilon'/k)$ on each subinterval. In Step 2 the algorithm draws a sample of $\tilde{O}(k/\varepsilon^2)$ points from $p$ and uses them to define an empirical distribution $\widehat{p}_m$. This is the only step in which points are drawn from $p$. For the rest of this intuitive explanation we pretend that the weight $\widehat{p}(I)$ that the empirical distribution $\widehat{p}_m$ assigns to each interval $I$ is actually the same as the true weight $p(I)$ (Lemma 3.1 below shows that this is not too far from the truth).

Before continuing with our explanation of the algorithm, let us digress briefly by imagining for a moment that the target distribution $p$ actually is a $k$-flat distribution (i.e., that $\mathrm{opt}_k(p) = 0$). In this case there are at most $k$ "breakpoints", and hence at most $k$ intervals $I_j$ for which $\alpha_{\widehat{p}_m}(I_j, I_{j+1}) > 0$, so computing the $\alpha_{\widehat{p}_m}(I_j, I_{j+1})$ values would be an easy way to identify the true breakpoints (and given these it is not difficult to construct a high-accuracy hypothesis).

In reality, we may of course have $\mathrm{opt}_k(p) > 0$; this means that if we try to use the $\alpha_{\widehat{p}_m}(I_j, I_{j+1})$ criterion to identify "breakpoints" of the optimal $k$-flat distribution that is closest to $p$ (call this $k$-flat distribution $q$), we may sometimes be "fooled" into thinking that $q$ has a breakpoint in an interval $I_j$ where it does not (but rather the value $\alpha_{\widehat{p}_m}(I_j, I_{j+1})$ is large because of the difference between $q$ and $p$). However, recall that by assumption we have $\mathrm{opt}_k(p) \leq \varepsilon$; this bound can be used to show that there cannot be too many intervals $I_j$ for which a large value of $\alpha_{\widehat{p}_m}(I_j, I_{j+1})$ suggests a "spurious breakpoint" (see the proof of Lemma 3.3). This is helpful, but in and of itself not enough; since our partition $I_1, \ldots, I_z$ divides $[0, 1)$ into $k/\varepsilon'$ intervals, a naive approach based on this would result in a $(k/\varepsilon')$-flat hypothesis distribution, which in turn would necessitate a sample complexity of $\tilde{O}(k/\varepsilon'^3)$, which is unacceptably high. Instead, our algorithm performs a careful process of iteratively merging consecutive intervals for which the $\alpha_{\widehat{p}_m}(I_j, I_{j+1})$ criterion indicates that a merge will not adversely affect the final accuracy by too much. As a result of this process we end up with $k \cdot \mathrm{polylog}(1/\varepsilon)$ intervals for the final hypothesis, which enables us to output a $(k \cdot \mathrm{polylog}(1/\varepsilon'))$-flat final hypothesis using $\tilde{O}(k/\varepsilon'^2)$ draws from $p$.

In more detail, this iterative merging is carried out by the main loop of the algorithm in Step 4. Going into the $t$-th iteration of the loop, the algorithm has a partition $\mathcal{P}_{t-1}$ of $[0, 1)$ into disjoint sub-intervals, and a set $\mathcal{F}_{t-1} \subseteq \mathcal{P}_{t-1}$ (i.e., every interval belonging to $\mathcal{F}_{t-1}$ also belongs to $\mathcal{P}_{t-1}$). Initially $\mathcal{P}_0$ contains all the intervals $I_1, \ldots, I_z$ and $\mathcal{F}_0$ is empty. Intuitively, the intervals in $\mathcal{P}_{t-1} \setminus$

$\mathcal{F}_{t-1}$ are still being "processed"; such an interval may possibly be merged with a consecutive interval from $\mathcal{P}_{t-1} \setminus \mathcal{F}_{t-1}$ if doing so would only incur a small "cost" (see condition (iii) of Step 4(b) of the algorithm). The intervals in $\mathcal{F}_{t-1}$ have been "frozen" and will not be altered or used subsequently in the algorithm.

### 3.1.2 The algorithm

---

**Algorithm** `Learn-WB-small-opt-`$k$`-histogram`:

**Input:** parameters $k \geq 1, \varepsilon > 0$; access to i.i.d. draws from target distribution $p$ over $[0, 1)$

**Output:** If (i) $p$ is $\frac{\varepsilon/\log(1/\varepsilon)}{384k}$-well-behaved and (ii) $\mathrm{opt}_k(p) \leq \varepsilon$, then with probability at least $99/100$ the output is a distribution $q$ such that $d_{\mathrm{TV}}(p, q) \leq 2\mathrm{opt}_k(p) + 3\varepsilon$.

1. Let $\varepsilon' = \varepsilon/\log(1/\varepsilon)$. Run Algorithm `Approximately-Equal-Partition` on input parameter $\frac{\varepsilon'}{6k}$ to partition $[0, 1)$ into $z = \Theta(k/\varepsilon')$ intervals $I_1 = [i_0, i_1), \ldots, I_z = [i_{z-1}, i_z)$, where $i_0 = 0$ and $i_z = 1$, such that with probability at least $99/100$, for each $j \in \{1, \ldots, z\}$ we have $p([i_{j-1}, i_j)) \in [\varepsilon'/12k, \varepsilon'/2k]$ (assuming $p$ is $\varepsilon'/(384k)$-well-behaved).

2. Draw $m = \tilde{O}(k/\varepsilon'^2)$ points from $p$ and let $\widehat{p}_m$ be the resulting empirical distribution.

3. Set $\mathcal{P}_0 = \{I_1, I_2, \ldots I_z\}$, and $\mathcal{F}_0 = \emptyset$.

4. Let $s = \log_2 \frac{1}{\varepsilon'}$. Repeat for $t = 1, \ldots$ until $t = s$:

   (a) Initialize $\mathcal{P}_t$ to $\emptyset$ and $\mathcal{F}_t$ to $\mathcal{F}_{t-1}$.

   (b) Without loss of generality, assume $\mathcal{P}_{t-1} = \{I_{t-1,1}, \ldots, I_{t-1,z_{t-1}}\}$ where interval $I_{t-1,i}$ is to the left of $I_{t-1,i+1}$ for all $i$. Scan left to right across the intervals in $\mathcal{P}_{t-1}$ (i.e., iterate over $i = 1, \ldots, z_{t-1} - 1$). If intervals $I_{t-1,i}, I_{t-1,i+1}$ are (i) both not in $\mathcal{F}_{t-1}$, and (ii) $\alpha_{\widehat{p}_m}(I_{t-1,i}, I_{t-1,i+1}) > \varepsilon'/(2k)$, then add both $I_{t-1,i}$ and $I_{t-1,i+1}$ into $\mathcal{F}_t$.

   (c) Initialize $i$ to 1, and repeatedly execute one of the following four (mutually exclusive and exhaustive) cases until $i > z_{t-1}$:
   [Case 1] $i \leq z_{t-1} - 1$ and $I_{t-1,i} = [a, b), I_{t-1,i+1} = [b, c)$ are consecutive intervals both not in $\mathcal{F}_t$. Add the merged interval $I_{t-1,i} \cup I_{t-1,i+1} = [a, c)$ into $\mathcal{P}_t$. Set $i \leftarrow i + 2$.
   [Case 2] $i \leq z_{t-1} - 1$ and $I_{t-1,i} \in \mathcal{F}_t$. Set $i \leftarrow i + 1$.
   [Case 3] $i \leq z_{t-1} - 1$, $I_{t-1,i} \notin \mathcal{F}_t$ and $I_{t-1,i+1} \in \mathcal{F}_t$. Add $I_{t-1,i}$ into $\mathcal{F}_t$ and set $i \leftarrow i + 2$.
   [Case 4] $i = z_{t-1}$. Add $I_{t-1,z_{t-1}}$ into $\mathcal{F}_t$ if $I_{t-1,z_{t-1}}$ is not in $\mathcal{F}_t$ and set $i \leftarrow i + 1$.

   (d) Set $\mathcal{P}_t \leftarrow \mathcal{P}_t \cup \mathcal{F}_t$.

5. Output the $|\mathcal{P}_s|$-flat hypothesis distribution $(\widehat{p}_m)^{\mathcal{P}_s}$.

---

### 3.1.3 Analysis of the algorithm and proof of Theorem 4

It is straightforward to verify the claimed running time given Lemma 2.1, which bounds the running time of `Approximately-Equal-Partition`. Indeed, we note that Step 2, which simply draws $\tilde{O}(k/\varepsilon'^2)$ points and constructs the resulting empirical distribution, dominates the overall running time. In the rest of this subsubsection we prove correctness.

We first observe that with high probability the empirical distribution $\widehat{p}_m$ defined in Step 2 gives a high-accuracy estimate of the true probability of any union of consecutive intervals from $I_1, \ldots, I_z$. The following lemma from [CDSS14] follows from the standard multiplicative Chernoff bound:

**Lemma 3.1** (Lemma 12, [CDSS14])**.** *With probability* $99/100$ *over the sample drawn in Step 2, for every* $0 \leq a < b \leq z$ *we have that* $|\widehat{p}_m([i_a, i_b)) - p([i_a, i_b))| \leq \sqrt{\varepsilon'(b - a)} \cdot \varepsilon'/(10k)$.

We henceforth assume that this $99/100$-likely event indeed takes place, so the above inequality holds for all $0 \leq a < b \leq z$. We use this to show that the $\alpha_{\widehat{p}_m}(I_{t-1,i}, I_{t-1,i+1})$ value that the algorithm

uses in Step 4(b) is a good proxy for the actual value $\alpha_p(I_{t-1,i}, I_{t-1,i+1})$ (which of course is not accessible to the algorithm):

**Lemma 3.2.** *Fix $1 \leq t \leq s$. Then we have $|\alpha_{\widehat{p}_m}(I_{t-1,i}, I_{t-1,i+1}) - \alpha_p(I_{t-1,i}, I_{t-1,i+1})| \leq 2\varepsilon'/(5k)$.*

Due to space constraints the proofs of all lemmas in this section are deferred to Appendix A.

For the rest of the analysis, let $q$ denote a fixed $k$-flat distribution that is closest to $p$, so $\|p - q\|_1 = \mathrm{opt}_k(p)$. (We note that while $\mathrm{opt}_k(p)$ is defined as $\inf_{q \in \mathcal{C}} \|p - q\|_1$, standard closure arguments can be used to show that the infimum is actually achieved by some $k$-flat distribution $q$.) Let $\mathcal{Q}$ be the partition of $[0, 1)$ corresponding to the intervals on which $q$ is piecewise constant. We say that a *breakpoint* of $\mathcal{Q}$ is a value in $[0, 1]$ that is an endpoint of one of the (at most) $k$ intervals in $\mathcal{Q}$.

The following important lemma bounds the number of intervals in the final partition $\mathcal{P}_s$:

**Lemma 3.3.** *$\mathcal{P}_s$ contains at most $O(k \log^2(1/\varepsilon))$ intervals.*

The following definition will be useful:

**Definition 5.** Let $\mathcal{P}$ denote any partition of $[0, 1)$. We say that partition $\mathcal{P}$ is $\varepsilon'$-*good* for $(p, q)$ if for every breakpoint $v$ of $\mathcal{Q}$, the interval $I$ in $\mathcal{P}$ containing $v$ satisfies $p(I) \leq \varepsilon'/(2k)$.

The above definition is justified by the following lemma:

**Lemma 3.4.** *If $\mathcal{P}$ is $\varepsilon'$-good for $(p, q)$, then $\|p - (p)^{\mathcal{P}}\|_1 \leq 2\mathrm{opt}_k(p) + \varepsilon'$.*

We are now in a position to prove the following:

**Lemma 3.5.** *There exists a partition $\mathcal{R}$ of $[0, 1)$ that is $\varepsilon'$-good for $(p, q)$ and satisfies*

$$\|(p)^{\mathcal{P}_s} - (p)^{\mathcal{R}}\|_1 \leq \varepsilon.$$

We construct the claimed $\mathcal{R}$ based on $\mathcal{P}_s, \mathcal{P}_{s-1}, \ldots, \mathcal{P}_0$ as follows: (i) If $I$ is an interval in $\mathcal{P}_s$ not containing a breakpoint of $\mathcal{Q}$, then $I$ is also in $\mathcal{R}$; (ii) If $I$ is an interval in $\mathcal{P}_s$ that does contain a breakpoint of $\mathcal{Q}$, then we further partition $I$ into a set of intervals $S$ in a recursive manner using $\mathcal{P}_{s-1}, \ldots, \mathcal{P}_0$ (see Appendix A.4). Finally, by putting everything together we can prove Theorem 4:

***Proof of Theorem 4.*** By Lemma 3.4 applied to $\mathcal{R}$, we have that $\|p - (p)^{\mathcal{R}}\|_1 \leq 2\mathrm{opt}_k(p) + \varepsilon'$. By Lemma 3.5, we have that $\|(p)^{\mathcal{P}_s} - (p)^{\mathcal{R}}\|_1 \leq \varepsilon$; thus the triangle inequality gives that $\|p - (p)^{\mathcal{P}_s}\|_1 \leq 2\mathrm{opt}_k(p) + 2\varepsilon$. By Lemma 3.3 the partition $\mathcal{P}_s$ contains at most $O(k \log^2(1/\varepsilon))$ intervals, so both $(p)^{\mathcal{P}_s}$ and $(\widehat{p}_m)^{\mathcal{P}_s}$ are $O(k \log^2(1/\varepsilon))$-flat distributions. Thus, $\|(p)^{\mathcal{P}_s} - (\widehat{p}_m)^{\mathcal{P}_s}\|_1 = \|(p)^{\mathcal{P}_s} - (\widehat{p}_m)^{\mathcal{P}_s}\|_{\mathcal{A}_\ell}$, where $\ell = O(k \log^2(1/\varepsilon))$ and $\mathcal{A}_\ell$ is the family of all subsets of $[0, 1)$ that consist of unions of up to $\ell$ intervals (which has VC dimension $2\ell$). Consequently by the VC inequality (Theorem 3, for a suitable choice of $m = \tilde{O}(k/\varepsilon'^2)$), we have that $\mathbf{E}[\|(p)^{\mathcal{P}_s} - (\widehat{p}_m)^{\mathcal{P}_s}\|_1] \leq 4\varepsilon'/100$. Markov's inequality now gives that with probability at least $96/100$, we have $\|(p)^{\mathcal{P}_s} - (\widehat{p}_m)^{\mathcal{P}_s}\|_1 \leq \varepsilon'$. Hence, with overall probability at least $19/20$ (recall the $1/100$ error probability incurred in Lemma 3.1), we have that $\|p - (\widehat{p}_m)^{\mathcal{P}_s}\|_1 \leq 2\mathrm{opt}_k(p) + 3\varepsilon$, and the theorem is proved. $\square$

### 3.2 A general reduction to the case of small opt for semi-agnostic learning

In this section we show that under mild conditions, the general problem of agnostic distribution learning for a class $\mathcal{C}$ can be efficiently reduced to the special case when $\mathrm{opt}_\mathcal{C}$ is not too large compared with $\varepsilon$. While the reduction is simple and generic, we have not previously encountered it in the literature on density estimation, so we provide a proof in Appendix A.5. A precise statement of the reduction follows:

**Theorem 6.** *Let $A$ be an algorithm with the following behavior: $A$ is given as input i.i.d. points drawn from $p$ and a parameter $\varepsilon > 0$. $A$ uses $m(\varepsilon) = \Omega(1/\varepsilon)$ draws from $p$, runs in time $t(\varepsilon) = \Omega(1/\varepsilon)$, and satisfies the following: if $\mathrm{opt}_\mathcal{C}(p) \leq 10\varepsilon$, then with probability at least $19/20$ it outputs a hypothesis distribution $q$ such that (i) $\|p - q\|_1 \leq \alpha \cdot \mathrm{opt}_\mathcal{C}(p) + \varepsilon$, where $\alpha$ is an absolute constant, and (ii) given any $r \in [0, 1)$, the value $q(r)$ of the pdf of $q$ at $r$ can be efficiently computed in $T$ time steps.*

*Then there is an algorithm $A'$ with the following performance guarantee: $A'$ is given as input i.i.d. draws from $p$ and a parameter $\varepsilon > 0$.[2] Algorithm $A'$ uses $O(m(\varepsilon/10) + \log\log(1/\varepsilon)/\varepsilon^2)$ draws from $p$, runs in time $O(t(\varepsilon/10)) + T \cdot \tilde{O}(1/\varepsilon^2)$, and outputs a hypothesis distribution $q'$ such that with probability at least $39/40$ we have $\|p - q'\|_1 \le 10(\alpha+2) \cdot \mathrm{opt}_{\mathcal{C}}(p) + \varepsilon$.*

### 3.3 Dealing with distributions that are not well behaved

The assumption that the target distribution $p$ is $\tilde{\Theta}(\varepsilon/k)$-well-behaved can be straightforwardly removed by following the approach in Section 3.6 of [CDSS14]. That paper presents a simple linear-time sampling-based procedure, using $\tilde{O}(k/\varepsilon)$ samples, that with high probability identifies all the "heavy" elements (atoms which cause $p$ to not be well-behaved, if any such points exist).

Our overall algorithm first runs this procedure to find the set $S$ of "heavy" elements, and then runs the algorithm presented above (which succeeds for well-behaved distributions, i.e., distributions that have no "heavy" elements) using as its target distribution the conditional distribution of $p$ over $[0,1) \setminus S$ (let us denote this conditional distribution by $p'$). A straightforward analysis given in [CDSS14] shows that (i) $\mathrm{opt}_k(p) \ge \mathrm{opt}_k(p')$, and moreover (ii) $d_{\mathrm{TV}}(p, p') \le \mathrm{opt}_k(p)$. Thus, by the triangle inequality, any hypothesis $h$ satisfying $d_{\mathrm{TV}}(h, p') \le C\mathrm{opt}_k(p') + \varepsilon$ will also satisfy $d_{\mathrm{TV}}(h, p) \le (C+1)\mathrm{opt}_k(p) + \varepsilon$ as desired.

## 4 Lower bounds on agnostic learning

In this section we establish that $\alpha$-agnostic learning with $\alpha < 2$ is information theoretically impossible, thus establishing Theorem 2.

Fix any $0 < t < 1/2$. We define a probability distribution $\mathcal{D}_t$ over a finite set of discrete distributions over the domain $[2N] = \{1, \ldots, 2N\}$ as follows. (We assume without loss of generality below that $t$ is rational and that $tN$ is an integer.) A draw of $p_{S_1, S_2, t}$ from $\mathcal{D}_t$ is obtained as follows.

1.  A set $S_1 \subset [N]$ is chosen uniformly at random from all subsets of $[N]$ that contain precisely $tN$ elements. For $i \in [N]$, the distribution $p_{S_1, S_2, t}$ assigns probability weight as follows:

    $$p_{S_1, S_2, t}(i) = \frac{1}{4N} \text{ if } i \in S_1, \qquad p_{S_1, S_2, t}(i) = \frac{1}{2N}\left(1 + \frac{t}{2(1-t)}\right) \text{ if } i \in [N] \setminus S_1.$$

2.  A set $S_2 \subset [N+1, \ldots, 2N]$ is chosen uniformly at random from all subsets of $[N+1, \ldots, 2N]$ that contain precisely $tN$ elements. For $i \in [N+1, \ldots, 2N]$, the distribution $p_{S_1, S_2, t}$ assigns probability weight as follows:

    $$p_{S_1, S_2, t}(i) = \frac{3}{4N} \text{ if } i \in S_2, \qquad \frac{1}{2N}\left(1 - \frac{t}{2(1-t)}\right) \text{ if } i \in [N] \setminus S_1.$$

Using a birthday paradox type argument, we show that no $o(\sqrt{N})$-sample algorithm can successfully distinguish between a distribution $p_{S_1, S_2, t} \sim \mathcal{D}_t$ and the uniform distribution over $[2N]$. We then leverage this indistinguishability to show that any $(2 - \delta)$-semi-agnostic learning algorithm, even for 2-flat distributions, must use a sample of size $\Omega(\sqrt{N})$ (see Appendix B for these proofs):

**Theorem 7.** *Fix any $\delta > 0$ and any function $f(\cdot)$. There is no algorithm $A$ with the following property: given $\varepsilon > 0$ and access to independent points drawn from an unknown distribution $p$ over $[2N]$, algorithm $A$ makes $o(\sqrt{N}) \cdot f(\varepsilon)$ draws from $p$ and with probability at least $51/100$ outputs a hypothesis distribution $h$ over $[2N]$ satisfying $\|h - p\|_1 \le (2 - \delta)\mathrm{opt}_2(p) + \varepsilon$.*

As described in the Introduction, via the obvious correspondence that maps distributions over $[N]$ to distributions over $[0,1)$, we get the following:

**Corollary 4.1.** *Fix any $\delta > 0$ and any function $f(\cdot)$. There is no algorithm $A$ with the following property: given $\varepsilon > 0$ and access to independent draws from an unknown distribution $p$ over $[0,1)$, algorithm $A$ makes $f(\varepsilon)$ draws from $p$ and with probability at least $51/100$ outputs a hypothesis distribution $h$ over $[0,1)$ satisfying $\|h - p\|_1 \le (2 - \delta)\mathrm{opt}_2(p) + \varepsilon$.*

## Footnotes

[1] We remark that our model is essentially equivalent to the "minimax rate of convergence under the $L_1$ distance" in statistics [DL01], and our results carry over to this setting as well.

[2] Note that now there is no guarantee that $\mathrm{opt}_{\mathcal{C}}(p) \le \varepsilon$; indeed, the point here is that $\mathrm{opt}_{\mathcal{C}}(p)$ may be arbitrary.

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
