[Supplementary Material · camera-ready-appendix.pdf]

# Appendix

## A  Omitted Proofs from Section 3

### A.1  Proof of Lemma 3.2

Observe that in iteration $t$, two consecutive intervals $I_{t-1,i}$ and $I_{t-1,i+1}$ correspond to two unions of consecutive intervals $I_a \cup \cdots \cup I_b$ and $I_{b+1} \cup \cdots \cup I_c$ respectively from the original partition $\mathcal{P}_0$. Moreover, since each interval in $\mathcal{P}_{t-1} \setminus \mathcal{F}_{t-1}$, $t > 1$, is formed by merging two consecutive intervals from $\mathcal{P}_{t-2} \setminus \mathcal{F}_{t-2}$, it must be the case that $b - a + 1, c - b + 1 \leq 2^{t-1} < 2^{s-1} \leq 1/(2\varepsilon')$. Hence, by Lemma 3.1, we have

$$|p(I_{t-1,i}) - \widehat{p}_m(I_{t-1,i}))| \leq \sqrt{\varepsilon' \cdot 2^{s-1}} \cdot \frac{\varepsilon'}{10k} \leq \frac{\varepsilon'}{10\sqrt{2}k}$$

and similarly,

$$|p(I_{t-1,i+1}) - \widehat{p}_m(I_{t-1,i+1}))| \leq \frac{\varepsilon'}{10\sqrt{2}k}.$$

To simplify notation, let $I = I_{t-1,i}$ and $J = I_{t-1,i+1}$. By definition of $\alpha$,

$$
\begin{aligned}
\alpha_p(I, J) &= \left| \frac{p(I)}{|I|} - \frac{p(I) + p(J)}{|I| + |J|} \right| |I| + \left| \frac{p(J)}{|J|} - \frac{p(I) + p(J)}{|I| + |J|} \right| |J| \\
&= \frac{2}{|I| + |J|} \big| p(I)|J| - p(J)|I| \big|.
\end{aligned}
\tag{1}
$$

A straightforward calculation now gives that

$$
\begin{aligned}
|\alpha_p(I, J) - \alpha_{\widehat{p}_m}(I, J)| &= \frac{2}{|I| + |J|} \Big| \big| p(I)|J| - p(J)|I| \big| - \big| \widehat{p}_m(I)|J| - \widehat{p}_m(J)|I| \big| \Big| \\
&\leq \frac{2}{|I| + |J|} \Big( \big| p(I) - \widehat{p}_m(I) \big| |J| + \big| p(J) - \widehat{p}_m(J) \big| |I| \Big) \\
&\leq 2\varepsilon'/(5k).
\end{aligned}
$$

### A.2  Proof of Lemma 3.3

We start by recording a basic fact that will be useful in the proof of the lemma. Let $p$ be a distribution over an interval $I$ and let $q$ be any sub-distribution over $I$. Perhaps contrary to initial intuition, the optimal scaling $c \cdot q$, $c > 0$, of $q$ to approximate $p$ (with respect to the $L_1$-distance) is not necessarily obtained by scaling $q$ so that $c \cdot q$ is a distribution over $I$. However, a simple argument (see e.g., Appendix A.1 of [CDSS14]) shows that scaling so that $c \cdot q$ is a distribution cannot result in $L_1$-error more than twice that of the optimal scaling:

**Claim A.1.** *Let $p, g : I \to \mathbb{R}^{\geq 0}$ be probability distributions over $I$ (so $\int_I p(x)dx = \int_I g(x)dx = 1$). Then, writing $\|f\|_1$ to denote $\int_I |f(x)|dx$, for every $a > 0$ we have that $\|p - g\|_1 \leq 2\|p - ag\|_1$.*

We now proceed with the proof of Lemma 3.3.

We first show that a total of at most $O(k \log(1/\varepsilon'))$ intervals are ever added into $\mathcal{F}_t$ across all executions of Step 4(b).

Suppose that intervals $I_{t-1,i}, I_{t-1,i+1}$ are added into $\mathcal{F}_t$ in some execution of Step 4(b). We consider the following two cases:

**Case 1:** $I_{t-1,i} \cup I_{t-1,i+1}$ contains at least one breakpoint of $\mathcal{Q}$. Since $\mathcal{Q}$ has at most $k$ breakpoints, this can happen at most $k$ times in total.

**Case 2:** $I_{t-1,i} \cup I_{t-1,i+1}$ does not contain any breakpoint of $\mathcal{Q}$. Then $I_{t-1,i} \cup I_{t-1,i+1}$ is a subset of an interval in $\mathcal{Q}$. Recalling that intervals $I_{t-1,i}, I_{t-1,i+1}$ were added into $\mathcal{F}_t$ in an execution of Step 4(b), we have that $\alpha_{\widehat{p}_m}(I_{t-1,i}, I_{t-1,i+1}) > \varepsilon'/(2k)$, and hence by Lemma 3.2, we have that $\alpha_p(I_{t-1,i}, I_{t-1,i+1}) \geq \frac{1}{5} \cdot \frac{\varepsilon'}{k}$. Claim A.1 now implies that the contribution to the

$L_1$ distance between $p$ and $q$ from $I_{t-1,i} \cup I_{t-1,i+1}$, i.e., $\int_{I_{t-1,i} \cup I_{t-1,i+1}} |p(x) - q(x)| dx$, is at least $\frac{1}{10} \frac{\varepsilon'}{k}$.

Since $\|p - q\|_1 = \mathrm{opt}_k(p)$, there can be at most

$$k + O\left(\frac{\mathrm{opt}_k(p) \cdot k}{\varepsilon'}\right) = O\left(k \cdot \log \frac{1}{\varepsilon}\right)$$

intervals ever added into $\mathcal{F}_t$ across all executions of Step 4(b) (note that for the last equality we have used the assumption that $\mathrm{opt}_k(p) \le \varepsilon$).

Next, we argue that each $\mathcal{F}_t$ satisfies $|\mathcal{F}_t| \le O(k \log^2(1/\varepsilon))$. We have bounded the number of intervals added into $\mathcal{F}_t$ in Step 4(b) by $O(k \log(1/\varepsilon'))$, so it remains to bound the number of intervals added in Step 4(c)(Case 3) and 4(c)(Case 4). It is clear that a total of at most $O(\log(1/\varepsilon'))$ intervals are ever added in 4(c)(Case 4). Inspection of Step 4(c)(Case 3) shows that for a given value of $t$, the number of intervals that this step adds to $\mathcal{F}_t$ is at most the number of "blocks" of consecutive $\mathcal{F}_t$-intervals. Since each interval added in Step 4(c)(Case 3) extends some blocks of consecutive $\mathcal{F}_t$-intervals but does not create a new one (and hence does not increase their number), across the $s = \log(1/\varepsilon')$ stages, the total number of intervals that can be added in executions of Step 4(c)(Case 3) is at most $O(k \log^2(1/\varepsilon'))$. It follows that we have $|\mathcal{F}_s| = O(k \log^2(1/\varepsilon))$ as claimed.

To bound $|\mathcal{P}_t \setminus \mathcal{F}_t|$, we observe that by inspection of the algorithm, for each $t$ we have $|\mathcal{P}_t \setminus \mathcal{F}_t| \le \frac{1}{2}|\mathcal{P}_{t-1} \setminus \mathcal{F}_{t-1}|$. Since $|\mathcal{P}_0| = \Theta(k/\varepsilon')$, it follows that $|\mathcal{P}_s \setminus \mathcal{F}_s| = O(k)$, and the lemma is proved.

### A.3   Proof of Lemma 3.4

Fix an interval $I$ in $\mathcal{P}$. If there does not exist an interval $J$ in $\mathcal{Q}$ such that $I \subseteq J$, then $I$ must contain a breakpoint of $\mathcal{Q}$, and hence since $\mathcal{P}$ is $\varepsilon'$-good for $(p, q)$, we have $p(I) \le \varepsilon'/(2k)$. This implies that the contribution to $\|(p)^{\mathcal{P}} - q\|_1$ that comes from $I$, namely $\int_I |(p)^{\mathcal{P}}(x) - q(x)| dx$, satisfies

$$
\begin{aligned}
\int_I |(p)^{\mathcal{P}}(x) - q(x)| dx &\le \int_I |(p)^{\mathcal{P}}(x) - p(x)| dx + \int_I |p(x) - q(x)| dx \\
&\le \int_I |p(x) - q(x)| dx + 2p(I) \\
&\le \int_I |p(x) - q(x)| dx + \frac{\varepsilon'}{k}.
\end{aligned}
$$

The other possibility is that there exists an interval $J$ in $\mathcal{Q}$ such that $I \subseteq J$. In this case, we have that

$$\int_I |(p)^{\mathcal{P}}(x) - q(x)| dx \le \int_I |p(x) - q(x)| dx.$$

Since there are at most $k$ intervals in $\mathcal{P}$ containing breakpoints of $\mathcal{Q}$, summing the above inequalities over all intervals $I$ in $\mathcal{P}$, we get that

$$\|(p)^{\mathcal{P}} - q\|_1 \le \|p - q\|_1 + \varepsilon' = \mathrm{opt}_k(p) + \varepsilon',$$

and hence

$$\|(p)^{\mathcal{P}} - p\|_1 \le \|(p)^{\mathcal{P}} - q\|_1 + \|p - q\|_1 \le 2\mathrm{opt}_k(p) + \varepsilon'.$$

### A.4   Proof of Lemma 3.5

We construct the claimed $\mathcal{R}$ based on $\mathcal{P}_s, \mathcal{P}_{s-1}, \ldots, \mathcal{P}_0$ as follows:

(i) If $I$ is an interval in $\mathcal{P}_s$ not containing a breakpoint of $\mathcal{Q}$, then $I$ is also in $\mathcal{R}$.

(ii) If $I$ is an interval in $\mathcal{P}_s$ that does contain a breakpoint of $\mathcal{Q}$, then we further partition $I$ into a set of intervals $S$ by calling procedure Refine-partition$(s, I)$. This recursive procedure exploits the local structure of the earlier, finer partitions $\mathcal{P}_{s-1}, \mathcal{P}_{s-2}, \ldots$ as described below.

---
**Procedure** `Refine-partition`:

**Input:** Integer $t$, Interval $J$

**Output:** $S$, a partition of interval $J$

1. If $t = 0$, then output $\{J\}$.
2. If $J$ is an interval in $\mathcal{P}_t$, then
   (a) If $J$ contains a breakpoint of $\mathcal{Q}$, then output `Refine-partition`$(t-1, J)$.
   (b) Otherwise output $\{J\}$.
3. Otherwise, $J$ is a union of two intervals in $\mathcal{P}_t$. Let $J_1$ and $J_2$ denote the two intervals in $\mathcal{P}_t$ such that $J_1 \cup J_2 = J$. Output `Refine-partition`$(t, J_1) \cup$ `Refine-partition`$(t, J_2)$.
---

We claim that $|\mathcal{R}|$ (the number of intervals in $\mathcal{R}$) is at most $|\mathcal{P}_s| + O(k \cdot \log \frac{1}{\varepsilon})$. To see this, note that each interval $I \in \mathcal{P}_s$ not containing a breakpoint of $\mathcal{Q}$ (corresponding to (i) above) translates directly to a single interval of $\mathcal{R}$. For each interval of type (ii) in $\mathcal{P}_s$, inspection of the `Refine-Partition` procedure shows that that these intervals are partitioned into at most $O(k \log(1/\varepsilon))$ intervals in $\mathcal{R}$.

In the rest of the proof, we show that for any interval $J$ in $\mathcal{P}_s$ containing at least one breakpoint of $\mathcal{Q}$, the contribution to the $L_1$ distance between $(p)^{\mathcal{P}_s}$ and $(p)^{\mathcal{R}}$ coming from interval $J$ is at most $|b_J| \cdot \frac{\varepsilon' \log \frac{1}{\varepsilon}}{k}$, where $b_J$ is the set of breakpoints of $\mathcal{Q}$ in $J$.

Consider a fixed breakpoint $v$ of $\mathcal{Q}$. Let $I_{t,v}$ denote the interval containing $v$ in the partition $\mathcal{P}_t$. If $I_{t,v}$ merges with another interval in $\mathcal{P}_t$ in Case 1 of Step 4(c), we denote that other interval as $I'_{t,v}$. Since $I_{t,v}$ merges with $I'_{t,v}$ in Case 1 of Step 4(c), these intervals are both not in $\mathcal{F}_t$ and hence were both not in $\mathcal{F}_{t-1}$ in Step 4(b). Consequently when $t > 1$ it must be the case that condition (ii) of Step 4(b) does not hold for these intervals, i.e. $\alpha_{\widehat{p}_m}(I_{t,v}, I'_{t,v}) \le \varepsilon'/(2k)$. It follows that by Lemma 3.2, we have that $\alpha_p(I_{t,v}, I'_{t,v})$ is at most $\frac{4\varepsilon'}{5k}$. When $t = 1$, we have a similar bound $\alpha_p(I_{t,v}, I'_{t,v}) \le \varepsilon'/k$, by using (1) and the fact that $p(I_{t,v}), p(I'_{t,v}) \le \varepsilon'/2k$ when $I_{t,v}, I'_{t,v} \in \mathcal{P}_0$.

On the other hand, inspection of the procedure `Refine-Partition` gives that if two intervals in $\mathcal{P}_t$ are unions of some intervals in `Refine-partition`$(s, I)$, and their union is an interval in $\mathcal{P}_{t+1}$, then there exists $v$ which is a breakpoint of $\mathcal{Q}$ such that the two intervals are $I_{t,v}$ and $I'_{t,v}$.

Thus, the contribution to the $L_1$ distance between $(p)^{\mathcal{P}_s}$ and $(p)^{\mathcal{R}}$ coming from interval $J$ is at most $\frac{\varepsilon'}{k} \cdot \log \frac{1}{\varepsilon'} \cdot |b_J|$. Summing over all intervals $J$ that contain at least one breakpoint and recalling that the total number of breakpoints is at most $k$, we get that the overall $L_1$ distance between $(p)^{\mathcal{P}_s}$ and $(p)^{\mathcal{R}}$ is at most $\varepsilon$.

## A.5 Proof of Theorem 6

*Proof.* The algorithm $A'$ works in two stages, which we describe and analyze below.

In the first stage, $A'$ iterates over $\lceil \log(20/\varepsilon) \rceil$ "guesses" for the value of $\mathrm{opt}_{\mathcal{C}}(p)$, where the $i$-th guess $g_i$ is $\frac{\varepsilon}{10} \cdot 2^{i-1}$ (so $g_1 = \frac{\varepsilon}{10}$ and $g_{\lceil \log(20/\varepsilon) \rceil} \ge 1$). For each value of $g_i$, it performs $r = O(1)$ runs of Algorithm $A$ (using a fresh sample from $p$ for each run) using parameter $g_i$ as the "$\varepsilon$" parameter for each run; let $h_{1,i}, \ldots, h_{r,i}$ be the $r$ hypotheses thus obtained for the $i$-th guess. It is clear that this stage uses $O(m(\varepsilon/10) + m(2\varepsilon/10) + \cdots) = O(m(\varepsilon))$ draws from $p$, and similarly that it runs in time $O(t(\varepsilon))$. If $\mathrm{opt}_{\mathcal{C}}(p) \le \varepsilon$, then (for a suitable choice of $r = O(1)$) we get that with probability at least $39/40$, some hypothesis $h_{1,\ell}$ satisfies $\|p - h_{1,\ell}\| \le \alpha \cdot \mathrm{opt}_{\mathcal{C}}(p) + \varepsilon/10$. Otherwise, there must be some $i \in \{2, \ldots, \lceil \log(20/\varepsilon) \rceil\}$ such that $g_i/2 < \mathrm{opt}_{\mathcal{C}}(p) \le g_i$; in this case, for a suitable choice of $r = O(1)$ we get that with probability at least $39/40$, there is some hypothesis $h_{i,\ell}$ that satisfies $\|p - h_{i,\ell}\|_1 \le \alpha \cdot \mathrm{opt}_{\mathcal{C}}(p) + g_i \le (\alpha + 2) \cdot \mathrm{opt}_{\mathcal{C}}(p)$. Thus in either event, with probability at least $39/40$ some $h_{i,\ell}$ satisfies $\|p - h_{i,\ell}\|_1 \le (\alpha + 2) \cdot \mathrm{opt}_{\mathcal{C}}(p) + \varepsilon/10$.

In the second stage, $A'$ runs a hypothesis selection procedure to choose one of the candidate hypotheses $h_{i,\ell}$. A number of such procedures are known (see e.g. Section 6.6 of [DL01] or

[DDS12, DK14, AJOS14]); all of them work by running some sort of "tournament" over the hypotheses, and all have the guarantee that with high probability they will output a hypothesis from the pool of candidates which has $L_1$ error (with respect to the target distribution $p$) not much worse than that of the best candidate in the pool. We use the classic Scheffé algorithm (see [DL01]) as described and analyzed in [AJOS14] (see Algorithm SCHEFFE* in Appendix B of that paper). Adapted to our context, this algorithm has the following performance guarantee:

**Proposition A.2.** *Let $p$ be a target distribution over $[0,1)$ and let $\mathcal{D}_\tau = \{p_j\}_{j=1}^N$ be a collection of $N$ distributions over $[0,1)$ with the property that there exists $i \in [N]$ such that $\|p - p_i\|_1 \leq \tau$. There is a procedure SCHEFFE which is given as input a parameter $\varepsilon > 0$ and a confidence parameter $\delta > 0$, and is provided with access to*

*(i) i.i.d. draws from $p$ and from $p_i$ for all $i \in [N]$, and*

*(ii) an evaluation oracle $eval_{p_i}$ for each $\in [N]$. This is a procedure which, on input $r \in [0,1)$, outputs the value $p_i(r)$ of the pdf of $p_i$ at the point $r$.*

*The procedure SCHEFFE has the following behavior: It makes $s = O\left((1/\varepsilon^2) \cdot (\log N + \log(1/\delta))\right)$ draws from $p$ and from each $p_i$, $i \in [N]$, and $O(s)$ calls to each oracle $eval_{p_i}$, $i \in [N]$, and performs $O(sN^2)$ arithmetic operations. With probability at least $1 - \delta$ it outputs an index $i^\star \in [N]$ that satisfies $\|p - p_{i^\star}\|_1 \leq 10\max\{\tau, \varepsilon\}$.*

The algorithm $A'$ runs the procedure SCHEFFE using the $N = O(\log(1/\varepsilon))$ hypotheses $h_{i,\ell}$, with its "$\varepsilon$" parameter set to $\frac{1}{10}\cdot$(the input parameter $\varepsilon$ that is given to $A'$) and its "$\delta$" parameter set to $1/40$. By Proposition A.2, with overall probability at least $19/20$ the output is a hypothesis $h_{i,\ell}$ satisfying $\|p - h_{i,\ell}\|_1 \leq 10(\alpha + 2)\mathrm{opt}_\mathcal{C}(p) + \varepsilon$. The overall running time and sample complexity are easily seen to be as claimed, and the theorem is proved. $\qquad\square$

# B    Proof of Theorem 7

We write $\mathcal{U}_{2N}$ to denote the uniform distribution over $[2N]$. The following proposition shows that $\mathcal{U}_{2N}$ has $L_1$ distance from $p_{S_1,S_2,t}$ almost twice that of the optimal 2-flat distribution:

**Proposition B.1.** *Fix any $0 < t < 1/2$.*

*1. For any distribution $p_{S_1,S_2,t}$ in the support of $\mathcal{D}_t$, we have*

$$\|\mathcal{U}_{2N} - p_{S_1,S_2,t}\|_1 = t.$$

*2. For any distribution $p_{S_1,S_2,t}$ in the support of $\mathcal{D}_t$, we have*

$$\mathrm{opt}_2(p_{S_1,S_2,t}) \leq \frac{t}{2}\left(1 + \frac{t}{1-t}\right).$$

*Proof.* Part (1.) is a simple calculation. For part (2.), consider the 2-flat distribution

$$q(i) = \begin{cases} \frac{1}{2N}\left(1 + \frac{t}{2(1-t)}\right) & \text{if } i \in [N] \\ \frac{1}{2N}\left(1 - \frac{t}{2(1-t)}\right) & \text{if } i \in [N+1,\ldots,2N] \end{cases}$$

It is straightforward to verify that $\|p_{S_1,S_2,t} - q\|_1 = \frac{t}{2}\left(1 + \frac{t}{1-t}\right)$ as claimed. $\qquad\square$

For a distribution $p$ we write $A^p$ to indicate that algorithm $A$ is given access to i.i.d. points drawn from $p$.

The following simple proposition states that no algorithm can successfully distinguish between a distribution $p_{S_1,S_2,t} \sim \mathcal{D}_t$ and $\mathcal{U}_{2N}$ using fewer than (essentially) $\sqrt{N}$ draws:

**Proposition B.2.** *There is an absolute constant $c > 0$ such that the following holds: Fix any $0 < t < 1/2$, and let $B$ be any "distinguishing algorithm" which receives $c\sqrt{N}$ i.i.d. draws from a distribution over $[2N]$ and outputs either "uniform" or "non-uniform". Then*

$$\left| \mathbf{Pr}[B^{\mathcal{U}_{[2N]}} \text{ outputs "uniform"}] - \mathbf{Pr}_{p_{S_1,S_2,t} \sim \mathcal{D}_t}[B^{p_{S_1,S_2,t}} \text{ outputs "uniform"}] \right| \le 0.01. \quad (2)$$

The proof is an easy consequence of the fact that in both cases (the distribution is $\mathcal{U}_{[2N]}$, or the distribution is $p_{S_1,S_2,t} \sim \mathcal{D}_t$), with probability at least 0.99 the $c\sqrt{N}$ draws received by $A$ are a uniform random set of $c\sqrt{N}$ distinct elements from $[2N]$ (this can be shown straighforwardly using a birthday paradox type argument).

Now we use Proposition B.2 to show that any $(2 - \delta)$-semi-agnostic learning algorithm even for 2-flat distributions must use a sample of size $\Omega(\sqrt{N})$, and thereby prove Theorem 7:

**Theorem 7.** *Fix any $\delta > 0$ and any function $f(\cdot)$. There is no algorithm $A$ with the following property: given $\varepsilon > 0$ and access to independent points drawn from an unknown distribution $p$ over $[2N]$, algorithm $A$ makes $o(\sqrt{N}) \cdot f(\varepsilon)$ draws from $p$ and with probability at least $51/100$ outputs a hypothesis distribution $h$ over $[2N]$ satisfying $\|h - p\|_1 \le (2 - \delta)\mathrm{opt}_2(p) + \varepsilon$.*

*Proof.* Fix a value of $\delta > 0$ and suppose, for the sake of contradiction, that there exists such an algorithm $A$. We describe how the existence of such an algorithm $A$ yields a distinguishing algorithm $B$ that violates Proposition B.2.

The algorithm $B$ works as follows, given access to i.i.d. draws from an unknown distribution $p$. It first runs algorithm $A$ with its "$\varepsilon$" parameter set to $\varepsilon := \frac{\delta^3}{12(2+\delta)}$, obtaining (with probability at least $51/100$) a hypothesis distribution $h$ over $[2N]$ such that $\|h - p\|_1 \le (2 - \delta)\mathrm{opt}_2(p) + \varepsilon$. It then computes the value $\|h - \mathcal{U}_{2N}\|_1$ of the $L_1$-distance between $h$ and the uniform distribution (note that this step uses no draws from the distribution). If $\|h - \mathcal{U}_{2N}\|_1 < 3\varepsilon/2$ then it outputs "uniform" and otherwise it outputs "non-uniform."

Since $\delta$ (and hence $\varepsilon$) is independent of $N$, the algorithm $B$ makes fewer than $c\sqrt{N}$ draws from $p$ (for $N$ sufficiently large). To see that the above-described algorithm $B$ violates (2), consider first the case that $p$ is $\mathcal{U}_{[2N]}$. In this case $\mathrm{opt}_2(p) = 0$ and so with probability at least $51/100$ the hypothesis $h$ satisfies $\|h - \mathcal{U}_{2N}\|_1 \le \varepsilon$, and hence algorithm $B$ outputs "uniform" with probability at least $51/100$.

On the other hand, suppose that $p = p_{S_1,S_2,t}$ is drawn from $\mathcal{D}_t$, where $t = \frac{\delta}{2+\delta}$. In this case, with probability at least $51/100$ the hypothesis $h$ satisfies

$$\|h - p_{S_1,S_2,t}\|_1 \le (2 - \delta)\mathrm{opt}_2(p_{S_1,S_2,t}) + \varepsilon \le (2 - \delta) \cdot \frac{t}{2} \cdot \left(1 + \frac{t}{1-t}\right) + \varepsilon,$$

by part (2.) of Proposition B.1. Since by part (1.) of Proposition B.1 we have $\|\mathcal{U}_{2N} - p_{S_1,S_2,t}\|_1 = t$, the triangle inequality gives that

$$\|h - \mathcal{U}_{2N}\|_1 \ge t - (2 - \delta) \cdot \frac{t}{2} \cdot \left(1 + \frac{t}{1-t}\right) - \varepsilon = 2\varepsilon,$$

where to obtain the final equality we recalled the settings $\varepsilon = \frac{\delta^3}{12(2+\delta)}$, $t = \frac{\delta}{2+\delta}$. Hence algorithm $B$ outputs "uniform" with probability at most $49/100$. Thus we have

$$\left| \mathbf{Pr}[B^{U_{[2N]}} \text{ outputs "uniform"}] - \mathbf{Pr}_{p_{S_1,S_2,t} \sim \mathcal{D}_t}[B^{p_{S_1,S_2,t}} \text{ outputs "uniform"}] \right| \ge 0.02$$

which contradicts (2) and proves the theorem. $\qquad \square$