[Reviews · NeurIPS 2014]

Submitted by Assigned_Reviewer_8

Near-optimal density estimation in near-linear time using variable-width histograms

Given a univariate distribution with support on the unit interval, this paper develops a method for estimating the density of the distribution via a histogram. The histogram has the following features: the bins are data-dependent, the bins are determined efficiently (near linear time), the number of bins k is specified by the user, and the estimator satisfies a "probably approximately correct" type performance guarantee. The paper is entirely theoretical. Pros: The writing is clear. I believe the results to be novel and theoretically pretty interesting. Cons: Some experiments would not have hurt the paper. A drawback of the theoretical framework is that model order k is not adaptively chosen.

My main quibble is with the omission of some relevant references. In particular, the algorithm developed by this paper has a course-to-fine kind of strategy, and harkens to multiscale and wavelet-based methods for density estimation. Here are two relevant references:

Rebecca Willett and Robert Nowak, "Multiscale Poisson Intensity and Density Estimation" IEEE Transactions on Information Theory, vol. 53, no. 9, pp. 3171-3187, 2007.

J. Klemelä. (2009). Multivariate histograms with data-dependent partitions. Statistica Sinica. 19(1): 159-176.

I would recommend the authors do both forward and reverse citation searches starting with these two papers. While this literature does not express its results in the PAC setting, and there is no doubt the above works pursue a different computational strategy, I think there are nonetheless some strong similarities. In fact, the above papers go beyond the submitted paper in looking at piecewise polynomial estimates (for smoothness classes beyond piecewise constant), adaptivity to model order, multiple dimensions, all while attaining optimal performance with efficient algorithms. I don't know if those results can be translated to give an apples-to-apples comparison with the submitted work, but if so, I would find that really interesting.

Other comments:

You say that you are restricting to the unit interval "without loss of generality." Yet, it's not clear whether your technique extends to densities with unbounded support.

At line 163, what do you mean that "the L_1 distance is a stronger metric". By Holder's inequality, and since your domain is the unit interval, we know that the L1 norm is bounded by the L2 norm, but wouldn't you need to reverse direction for your statement to make sense? I'm possibly forgetting some basic fact here.

The first paragraph of the "preliminaries" section confuses me on one point. You say that "for convenience in the rest of the paper we work with the L_1 distance." Yet you are also allowing distributions with point masses, in which case the distribution does not have a Lebesgue density. If you are allowing point masses, I think you need to stick to TV, and when you say density, be clear about the dominating measure. If you are allowing mixed continuous-discrete distributions, the term "density" seems inappropriate. You're estimating the probability measure.

The term "semi-agnostic learning" in the title of subsection 3.2 is not defined.

In the last line of the statement of theorem 6, I think you mean q' instead of q.

Summary: Novel interesting work on a theoretical problem in histogram density estimation. Missing some important references.

Submitted by Assigned_Reviewer_19

The paper considers the following basic problem. Given samples from an unknown distribution over [0,1], obtain its best $k$-histogram estimate. The main contribution of this paper is to provide a near sample-optimal agnostic algorithm that also runs in time linear in the number of samples, and therefore has nearly optimal running time as well. In particular, given input parameters $k$, and $\epsilon$ and samples from a distribution $p$, the algorithm with sample and time complexity $\tilde{O}(k/\epsilon^2)$, outputs a distribution $q$, which is a $k\log^2(1/\epsilon)$ histogram and satisfies, $d(p,q) < opt(p)+\epsilon$, where $d(.,.)$ is the total variation distance, and $opt(p)$ is the smallest total variation distance of $p$ from a $k$-histogram distribution.

As the authors mention, there is a brute force algorithm that requires the optimal $O(k/\epsilon^2)$ samples, and runs in time exponential in $k$, and $1/\epsilon$, and the more recent works still require $O(k^3)$ time using dynamic programming and linear programming. In this work the authors avoid these methods altogether, and approach the problem in a more direct way. They first obtain a ``fine'' histogram with k\poly(1/\epsilon) steps and then perform a clever merging of the steps to obtain the required number of steps. It is surprising, and extremely interesting solution to a simple and interesting problem. The paper is well motivated and written clearly.

The authors provide an additional multiplicative $\log^2(1/\epsilon)$ steps. Is there any way to reduce this number, say to something like $O(k)+f(\epsilon)$. They can comment about the difficulty that potentially arises in obtaining fewer steps with similar guarantees.

I don't understand the comment in line 156-157 that ``MLE does not exist for very simple classes of distributions''.
Summary: I strongly recommend the acceptance of this paper. I think this kind of research, and the algorithmic flavor has good potential in the learning community.

Submitted by Assigned_Reviewer_30

This paper tackles a fundamental problem in density estimation: given independent draws from a probability distribution, how can one estimate the underlying density? Exact recovery is impossible, so various heuristics are often used, eg kernel density estimators.

This paper takes a more learning-theoretic approach and develops the following approach. Given a parameter k, it will produce a k-histogram that approximates the best k-histogram (and therefore the target, assuming the best k-histogram approximates the target well) in linear time. The approximation factor is ~1 an this is about as good as possible.

The algorithm is given k/epsilon^2 samples, which is known to be sufficient, but trying all histograms is not efficient. The best current solution takes k^3 time, and this paper reduces it to linear. The algorithm draws samples, partitions into too many intervals, and iteratively merges contiguous intervals without causing too many problems. The algorithm is nicely described and seems correct.

The problem is important and the result beats the state of the art. Histograms are a natural set to compete against, so this would make a welcome addition to NIPS.
Summary: Nice advance in density estimation. Basically optimal for approximating k-histograms. A nice paper, should be accepted.
Author Feedback
Author rebuttal: We thank all of the reviewers for their thoughtful comments. We address
comments and questions from the various reviewers in turn below.

Reviewer (1):

* It is an interesting question whether the number of "steps"
in the final hypothesis can be reduced from O(k*log^2(1/eps)) to O(k)+f(eps);
we will think about whether the analysis establishing Lemma 3.3 can
be strengthened towards this end.

* Regarding the MLE: see the first full paragraph on p. 971 of

[Bir97] L. Birge. Estimation of unimodal densities without smoothness
assumptions. Annals of Statistics, 25(3):970--981, 1997.

Quoting, "...the MLE over the family of all unimodal densities does not
exist any more [when the mode is unknown] because it tends to put an infinite
density at one of the observations."

More discussion on non-existence of MLEs can be found at

http://math.stackexchange.com/questions/85782/when-does-a-maximum-likelihood-estimate-fail-to-exist

and the references therein. (If the paper is accepted we may just
remove the confusing sentence on lines 156-157 since space constraints
will probably preclude a detailed explanation.)

Reviewer (3):

* We thank the reviewer for identifying these references,
which indeed seem relevant. We will do a detailed literature search
as suggested and incorporate its results into the final version
if the paper is accepted. We would like to point that the results
of these references differ from our work in a number of ways,
and in particular are incomparable to our main result. More specifically,
the notion of agnostic learning, which is crucial for us,
is not considered in these references. Moreover, our paper differs from this related work in its emphasis on having an extremely computationally efficient algorithm, running in essentially linear time in the data.

* We will clarify that our restriction to the unit interval is w.l.o.g.
for densities with bounded support.

* Lines 163--164: in this context the L_2 distance between densities
p,q over [N] (where \sum_{i=1}^N p(i)=1 and likewise for q) is

L_2(p,q) = \sqrt{sum_i (p(i)-q(i))^2}

while the L_1 norm is

L_1(p,q) = \sum_i |p(i)-q(i)|

The latter is always at least as large as the former, so achieving
epsilon-accuracy for L_2 is easier than achieving eps-accuracy for L_1.

* First paragraph of preliminaries: we are allowing point masses
(to deal with as broad a class of distributions as possible). We will
stick with TV rather than L_1, and change terminology to "measure"
rather than "density" throughout. (Note that for comparison with the
[ILR12] results, in the previous bullet, this is not an issue
since those are distributions over the finite set [N].)

* We will define "semi-agnostic" so the title of Section 3.2 and other
usage of the phrase "semi-agnostic" is clear (will change lines 89-90)

* Line 376: indeed this should be q', we will change this.